# Triad Constraints for Learning Causal Structure of Latent Variables

**Ruichu Cai**[*1], **Feng Xie** [*1], **Clark Glymour** [2], **Zhifeng Hao** [1,3], **Kun Zhang** [2]

[1] School of Computer Science, Guangdong University of Technology, Guangzhou, China
[2] Department of Philosophy, Carnegie Mellon University, Pittsburgh, USA
[3] School of Mathematics and Big Data, Foshan University, Foshan, China
cairuichu@gdut.edu.cn,xiefeng009@gmail.com,cg09@andrew.cmu.edu
zfhao@gdut.edu.cn,kunz1@cmu.edu

## Abstract

Learning causal structure from observational data has attracted much attention, and it is notoriously challenging to find the underlying structure in the presence of confounders (hidden direct common causes of two variables). In this paper, by properly leveraging the non-Gaussianity of the data, we propose to estimate the structure over latent variables with the so-called Triad constraints: we design a form of "pseudo-residual" from three variables, and show that when causal relations are linear and noise terms are non-Gaussian, the causal direction between the latent variables for the three observed variables is identifiable by checking a certain kind of independence relationship. In other words, the Triad constraints help us to locate latent confounders and determine the causal direction between them. This goes far beyond the Tetrad constraints and reveals more information about the underlying structure from non-Gaussian data. Finally, based on the Triad constraints, we develop a two-step algorithm to learn the causal structure corresponding to measurement models. Experimental results on both synthetic and real data demonstrate the effectiveness and reliability of our method.

## 1 Introduction

Traditional methods for causal discovery, which aims to find causal relations from (purely) observational data, can be roughly divided into two categories, namely constraint-based methods including PC [Spirtes and Glymour, 1991] and FCI [Spirtes *et al.*, 1995; Colombo *et al.*, 2012], and score-based ones such as GES [Chickering, 2002] and GES with generalized scores [Huang *et al.*, 2018]. A number of methods focus on estimating causal relationships between observed variables and fail to recover the underlying causal structure of latent variables. For example, from large enough data generated by the structure in Figure 1, where $L_i$ are latent variables and $X_i$ are observed ones, we may only get a complete graph using the PC algorithm [Spirtes and Glymour, 1991], a widely-used constraint-based method, since there is no d-separation relation among the observed variables (although $\{X_1\}$ and $\{X_2, X_3\}$ are d-separated by $L_1$, which is latent). Besides, in reality we can measure only a limited number of variables and the causal influences may happen at the level of latent variables, so we are often concerned about the causal structure of latent variables; see e.g., Bartholomew *et al.* [2008].

There exist several methods for causal discovery in the case with confounders. Spirtes *et al.* [2000] attempt to resolve this problem using the so-called Tetrad constraints [Spearman, 1928]. Inspired by Tetrad constraints, various contributions have been made towards estimating structure over latent

---

variables. For instance, Silva and Scheines [2005] presented testable statistical conditions to identify d-separations in linear latent variable models, Silva *et al.* [2006] propose the BPC algorithm using Tetrad constraints to discovery causal structure of latent variables, and Shimizu *et al.* [2009] further applied analysis based on the Linear, Non-Gaussian, Acyclic Model (LiNGAM) [Shimizu *et al.*, 2006] to the recovered latent variables to further improve the estimated causal relations between them; Sullivant *et al.* [2010] showed that a sub-matrix of the covariance matrix with low rank corresponds to conditional independence constraints on the collections of Gaussian data and proposed a trek separation criterion to learn causal structure. Recently, Kummerfeld and Ramsey [2016] used the extended t-separation [Spirtes, 2013] to infer causal relations of latent variables, with the FindOneFactorClusters (FOFC) algorithm. However, these methods fail to work when latent variables have fewer than three pure measurement variables. Furthermore, even when this condition holds, Tetrad and its variants may not be able to find the causal direction between latent variables. Over-complete independent component analysis offers another method [Hoyer *et al.*, 2008], as an extension of the LiNGAM analysis; however, this analysis is generally hard to do, especially when there are relatively many latent variables, and the method does not focus on the structure of latent variables. More recently, Zhang *et al.* [2017] and Huang *et al.* [2015] deal with a specific type of confounders, which can be written as functions of the time/domain index in nonstationary/heterogeneous data. Overall, learning the structure of latent variables is a challenging problem; for instance, none of the above methods is able to recover the causal structure as shown in Figure 1.

It is desirable to develop testable conditions on the observed data to estimate the structure of latent variables. Interestingly, we find that given three variables in the *non-Gaussian* case, the independence condition between one of them and a certain linear combination of the remaining two variables gives hints as to the causal structure even in the presence of latent confounders. In particular, given a set of three distinct and dependent variables $\{X_i, X_j, X_k\}$, we define a particular type of "regression residual," $E_{(i,j\,|\,k)} := X_i - \frac{\text{Cov}(X_i, X_k)}{\text{Cov}(X_j, X_k)} \cdot X_j$. Then whether $E_{(i,j\,|\,k)}$ is independent from $X_k$ contains information regarding where latent confounders might be and the causal relationships among them. We term this condition the Triad constraint.

We further extend our Triad constraints to learn the structure of a wide class of linear latent structure models from non-Gaussian data. Specifically, we propose a two-phase algorithm to discover the causal relationships of latent variables. It first finds pure clusters (clusters of variables having only one common latent variable and no observed parent) from observed data in phase I. Then in phase II it learns the causal order of latent variables based on the clusters. Compared with Tetrad constraints, Triad constraints can reveal more information about the causal structure involving latent variables for non-Gaussian data. For instance, Triad

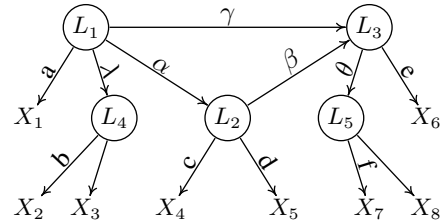

Figure 1: A causal structure involving 5 latent variables.

constraints can be used to locate the latent variables $L_i$, $i = 1, ..., 5$, in Figure 1 and identify their structure, including their causal direction, but Tetrad constraints cannot (see the details in Section 4).

Our main contributions include 1) proposing a novel constraint involving only three non-Gaussian variables, namely the Triad constraint, and showing the connection between this constraint and the underlying causal structure, which helps identify causal information of latent confounders, and 2) developing a two-phase algorithm to learn the causal structure of latent variables, including causal skeleton and causal directions, based on the Triad constraints.

## 2  Problem Definition

In this work, we focus on a particular type of linear latent structure model. Let $\mathbf{X} = \{X_1, X_2, ...X_m\}$ denote the observed variable set, $\mathbf{L} = \{L_1, L_2, ...L_n\}$ denote the latent variable set, and $\mathbf{V} = \mathbf{X} \cup \mathbf{L}$ denote the full variable set. In the linear latent structure model, the data generation process follows: 1) the structure of $\mathbf{V}$ can be represented by a Direct Acyclic Graph (DAG), 2) no observed variable in $\mathbf{X}$ is an ancestor of any latent variable in $\mathbf{L}$, 3) the generation of $\mathbf{V}$ is assumed to follow $V_i = \sum_{V_k \in Pa(V_i), k \neq i} b_{ik} V_k + \varepsilon_{V_i}, i = 1, 2, ..., m + n$, where $Pa(V_i)$ contains all the parent variables of $V_i$ and

$b_{ik}$ is the causal strength from $V_k$ to $V_i$; and 4) all $\varepsilon_{V_i}$ are noise (disturbance) variables which are independent with each other.

BPC, FOFC, and their variants [Silva *et al.*, 2006; Kummerfeld and Ramsey, 2016] have been shown to be able to recover a certain amount of causal information for some linear latent structure models from observed data. These methods usually assume that each latent variable has at least three pure measurement variables, which may not hold in practice, e.g., for the example given in Figure 1; furthermore, they cannot always recover the causal direction between latent variables. Here, pure measurement variables are defined as measured variables that have only one latent parent and no observed parent.

Here, we greatly relax the structural assumption of Tetrad; we consider the case where each latent variable has two or more pure variables as children, under the assumption of non-Gaussianity of the noise terms. Here, pure variables are the variables that may be latent or observed but have only one parent. The model is defined as follows.

**Definition 1** (Non-Gaussian Two-Pure Linear Latent Structure Model). *A linear latent structure model is called a Non-Gaussian Two-Pure (NG2P) linear latent structure model if it further satisfies the following three assumptions:*

1) *[Purity Assumption] there is no direct edges between the observed variables;*

2) *[Two-Pure Child Variable Assumption] each latent variable has at least two pure variables as children;*

3) *[Non-Gaussianity Assumption] the noise terms are non-Gaussian.*

One may wonder how restrictive the above assumptions are and how to interpret the result produced by our proposed method when the assumptions, especially assumption 1), are violated. We will discuss such issues in Section 5.

## 3 Triad Constraints: A Brief Formulation

We begin with the definition of Triad constraints, the independence relationship between the "pseudo-residual" and the observed variables. It is worth noting that there is some related work that also exploits similar concepts to "pseudo-residual", e.g., in the context of auxiliary variables (or instrumental variables)[Chen *et al.*, 2017] or pseudo-variable [Drton and Richardson, 2004], but to the best of our knowledge, it has not been realized that the independence property involving such pseudo-residuals reflects structural asymmetry of the latent variables.

**Definition 2** (Triad constraints). *Suppose $X_i$, $X_j$ and $X_k$ are distinct and correlated variables and that all noise variables are non-Gaussian. Define the pseudo-residual of $\{X_i, X_j\}$ relative to $X_k$, which is called a reference variable, as*

$$E_{(i,j\,|\,k)} := X_i - \frac{Cov(X_i, X_k)}{Cov(X_j, X_k)} \cdot X_j. \tag{1}$$

*We say that $\{X_i, X_j\}$ and $X_k$ satisfy Triad constraint if and only if $E_{(i,j\,|\,k)} \perp\!\!\!\perp X_k$, i.e., $\{X_i, X_j\}$ and $X_k$ violate the Triad constraint if and only if $E_{(i,j\,|\,k)} \not\perp\!\!\!\perp X_k$.*

The following two theorems show some interesting properties of the Triad constraints, which are further explored to discover the causal structure among the latent variables. We first aim at the identification of the causal direction of latent variables by analyzing the variables in the clusters. The following theorem shows the asymmetry between the latent variables in light of the Triad condition in the non-Gaussian case.

**Theorem 1.** *Let $L_a$ and $L_b$ be two directed connected latent variables without confounders and let $\{X_i\}$ and $\{X_j, X_k\}$ be their children, respectively. Then if $\{X_i, X_j\}$ and $X_k$ violate the Triad constraint, $L_a \rightarrow L_b$ holds. In other words, if the Triad condition is violated and the latent variables have no confounders, then the latent variable of the reference variable is a child of the other latent variable.*

The proof is given in the Supplementary Material, and it heavily relies on the Darmois-Skitovich Theorem Kagan *et al.* [1973], which essentially says that as long as two variables share any non-Gaussian, independent component, they cannot be statistically independent. The following example

shows that Triad constraints help find the causal direction between two latent variables from their pure clusters.

**Example 1.** *Consider the example in Figure 1, clusters $\{X_1\}$ and $\{X_4, X_5\}$ have corresponding latent variables $L_1$ and $L_2$, respectively. Because $L_1 \rightarrow L_2$ without a confounder, any Triad condition with any child of $L_2$ is violated, i.e., $E_{(1,4\,|\,5)} \not\perp\!\!\!\perp X_5$, and $E_{(1,5\,|\,4)} \not\perp\!\!\!\perp X_4$, but $E_{(4,5\,|\,1)} \perp\!\!\!\perp X_1$. This shows the asymmetry between $L_1$ and $L_2$, implied by the three observed variables.*

One might wonder whether we can make use of the Triad constraints in the Gaussian case to infer the causal direction between $L_1$ and $L_2$ in the above example. Unfortunately, one can show $E_{(1,2\,|\,3)} \perp\!\!\!\perp X_3$, $E_{(1,3\,|\,2)} \perp\!\!\!\perp X_2$ and $E_{(2,3\,|\,1)} \perp\!\!\!\perp X_1$ when the variables are *jointly Gaussian*, and thus the asymmetry between $L_1$ and $L_2$ disappears.

The second theorem is about the property of the clusters in terms of the Triad constraints. Here we say a set of observed variables is a ***cluster*** if these variables have the same latent variable as the parent. Intuitively, if such variables are pure variables, they are equivalent under the Triad constraints. For example, $X_2$ and $X_3$ in Figure 1 have the same constraints. Theorem 2 formalizes this property of clusters and gives the criterion for finding clusters.

**Theorem 2.** *Let $S$ be a correlated variable set. If $\forall X_i, X_j \in S$ and $\forall X_k \in \mathbf{X} \setminus S$, $\{X_i, X_j\}$ and $X_k$ satisfy the Triad constraints, then $S$ is a cluster.*

The proof is given in the Supplementary Material. The following example illuminates how the theorem can be used to distinguish the cluster of the variables.

**Example 2.** *Consider the example in Figure 1, for $\{X_4, X_5\}$, one may find $\{X_4, X_5\}$ and $X_i$ satisfy Triad constraint, where $i = 1, 2, 3, 6, 7, 8$, so $\{X_4, X_5\}$ is a cluster. But for $\{X_1, X_4\}$, $E_{(1,4\,|\,5)}$ is not independent of $X_5$, so $\{X_1, X_4\}$ is not a cluster.*

## 4 Triad Constraint-Based Causal Latent Structure Discovery

In this section, we extend the above results to estimate the NG2P linear latent structure. To this end, we propose a two-phase algorithm to **L**earn the **S**tructure of latent variables based on **T**riad **C**onstraints (LSTC). It firstly finds pure clusters from the observed data (phase I), and then it learns the structure of the latent variables behind these clusters (phase II).

### 4.1 Phase 1: Finding Clusters

Theorem 2 has paved the way to discover the clusters of the variables. It also enables us to use a cluster fusion-like method to discover the clusters of observed variables and latent variables that have already been found, i.e., we recursively find the clusters of variables and merge the overlapping clusters. Here we consider two practical issues involved in such a recursive fusion algorithm. The first is what clusters are to be merged, and the second is how to check whether Triad constraints involving latent variables hold given that they are hidden.

For the merge problem, we find that the overlapping clusters can be directly merged into one cluster. This is because the overlapping clusters have the same latent variable as the parent under the NG2P linear latent structure. The validity of the merge step is guaranteed by Proposition 1.

**Proposition 1.** *Let $C_1$ and $C_2$ be two clusters. If $C_1$ and $C_2$ are overlapping, $C_1$ and $C_2$ share the same latent parent.*

This proposition holds true because of the equivalence of the pure variables in terms of Triad constraints. In particular, as shown in Theorem 2, all variables in a cluster have the same Triad constraints.

After we find and merge clusters, we associate each cluster with a latent variable and, in fact, replace the variables in the cluster by the corresponding latent variable. We will then continue finding clusters and merging clusters. Since we replace variables in the same cluster with the associated latent variable, clearly subsequent Triad constraints to be checked may involve latent variables. How can we check such constraints without knowing the values of the latent variables? Thanks to the linearity assumption and the transitivity of linear causal relations, one can use its child to test the Triad constraints. Consider the example in Figure 1. Suppose we already found the cluster $\{X_2, X_3\}$

and associated it with a latent variable, say $L_4$. Then one can see that if only one variable in this cluster, say $X_2$, is kept (i.e., $X_3$ is removed), then any subsequent Triad constraint, e.g., that of $\{X_1, L_4\}$ and $X_5$, holds true if and only if $\{X_1, X_2\}$ and $X_5$ holds because $X_3$ is not in the variable set and $L_4$ and its only child, $X_2$, have the same Triad properties relative to any other remaining variable. That means, we can just use the observed variables of $X_2$ as the values of $L_4$ and ignore all the other variables in the same cluster for the purpose of checking Triad constraints.

Consideration of the above two issues directly leads to the following algorithm, which includes three main steps: 1) find the clusters according to Theorem 2; 2) merge the overlapping clusters according to Proposition 1; 3) introduce a new latent variable to represent a newly discovered cluster and use the values of an arbitrary variable in the cluster as the observed values of the latent variable for subsequent Triad condition checking. This procedure is illustrated with the following example.

---

**Algorithm 1** FindClusters

---

**Input:** Data set $\mathbf{X} = \{X_1, ..., X_m\}$
**Output:** Partial causal structure $\mathcal{G}$

1: Initialize $\mathbf{C} = \varnothing$, $\mathcal{G} = \varnothing$, $\mathbf{V} = \mathbf{X}$;
2: **repeat**
3:     **for** each $\{V_i, V_j\} \in \mathbf{V}$ **do**
4:       **if** $V_i$ and $V_j$ **then**
5:         **if** $E_{(i,j\,|\,k)} \perp\!\!\!\perp V_k$ holds for $\forall V_k \in \mathbf{V} \setminus \{V_i, V_j\}$ **then**
6:           $\mathbf{C} = \mathbf{C} \cup \{\{V_i, V_j\}\}$;
7:         **end if**
8:       **end if**
9:     **end for**
10:     Merge all the overlapping sets in $\mathbf{C}$.
11:     **for** each $S \in \mathbf{C}$ **do**
12:       Introduce a latent variable $L$ for $S$ and initialize $L$ with the value of any variable of $S$;
13:       $\mathbf{V} = (\mathbf{V} \setminus S) \cup \{L\}$;
14:       $\mathcal{G} = \mathcal{G} \cup \{L \to V_i | V_i \in S\}$;
15:     **end for**
16: **until** $\mathbf{V}$ contains only latent variables.
17: **Return:** $\mathcal{G}$

---

**Example 3.** *Consider the example in Figure 1. First, we find the clusters $\{X_2, X_3\}$, $\{X_4, X_5\}$, $\{X_7, X_8\}$ based on the Theorem 2 (line 3-8). Second, introduce $L_4, L_2$ and $L_5$ as the parents for $\{X_2, X_3\}, \{X_4, X_5\}, \{X_7, X_8\}$, respectively, whose values are set to those of $X_2, X_4$ and $X_7$, respectively. Third, we find the clusters $\{X_1, L_4\}, \{X_6, L_5\}$ on the updated $\mathbf{V}$ based on Theorem 2 (line 3-8). Fourth, introduce $L_1$ and $L_3$ as the parents of $\{X_1, L_4\}$ and $\{X_6, L_5\}$, respectively. Finally, we return the clusters of the variables in the form of partial graph as $\mathcal{G} = \{L_1 \to \{X_1, L_4\}, L_4 \to \{X_2, X_3\}, L_2 \to \{X_4, X_5\}, L_3 \to \{X_6, L_5\}$ and $L_5 \to \{X_7, X_8\}\}$.*

### 4.2 Phase 2: Learning the Structure of Latent Variables

Given the clusters discovered in the previous step, we aim to recover the structure among the root latent variables of each cluster. Due to the availability of various independence test methods for the latent variables, the causal order is the focus of this learning procedure. As an immediate extension of Theorem 1, the root latent variable can be identified by checking the Triad constraints, as stated in the following proposition.

**Proposition 2.** *Given a latent variable $L_r$ and its two children $\{V_i, V_j\}$, $L_r$ is a root latent variable if and only if $E_{(k,i|j)} \perp\!\!\!\perp V_j$ holds for each $V_k$, where $V_k$ is a child of any other latent variables.*

This proposition inspires us to use a recursive approach to discover the causal order; we recursively identify the root latent variable and update the data by removing the root variable's effect, until the causal order over all latent variables is determined. The key concern of such recursive approach is whether Proposition 2 still works on the updated data.

Fortunately, we find that there is still asymmetry implied by the Triad constraints if we update the data as follows: let $\{V_i, V_j\}$ be two pure variables of the root latent $L_r$,

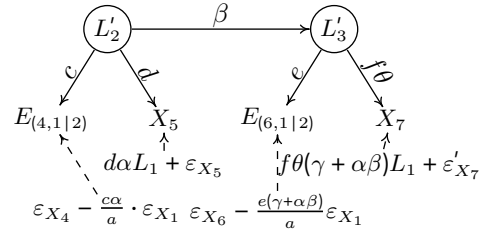

Figure 2: Structure obtained after removing the effects of $L_1$ through $\{X_1, X_2\}$, where $L'_2 = \varepsilon_{L_2}$, $L'_3 = e\beta\varepsilon_{L_2} + \varepsilon_{L_3}$, $\varepsilon'_{X_7} = f\varepsilon_{L_5} + \varepsilon_{X_7}$, and the influences of noise terms are shown by dashed lines.

for any other remaining latent variable $L$, we update the value of $V_k$, which is a child of the value of $L$, as $V_k := E_{(k,i|j)}$ and keep the value of the other children unchanged. On the updated data, the property of the root, i.e., $E_{(k,i|j)}$ is independent of $X_j$ still holds. Recall the example given in Figure 1, although such a removal step introduces common effect into the updated variables, i.e., $E_{(4,1|2)}$ and $E_{(6,1|2)}$ share a common noise $\varepsilon_{X_1}$, as seen in Figure 2, $\{E_{(4,1|2)}, \{E_{(6,1|2)}\}$ and $X_5$ satisfy the Triad constraint, while $\{E_{(4,1|2)}, \{E_{(6,1|2)}\}$ and $X_7$ violate it. More detail is given in the Supplementary Material.

Given the causal order of the variables, we can find the causal structure simply by removing redundant edges from the full acyclic graph using the independence test methods. Here we adopt the independence test method proposed in [Silva *et al.*, 2006] (see Theorem 19 therein for the detail). Finally, we present the following recursive algorithm for learning the structure over latent variables, and give the following example for illustration.

---

**Algorithm 2** LearnLatentStructure

---

**Input:** Partial causal structure $\mathcal{G}$
**Output:** Complete causal structure $\mathcal{G}$

1: Initialize $\mathcal{L}$ with the root variables of each subgraph in $\mathcal{G}$ and $\mathbf{L}_r = \phi$;
2: Select two pure child for each $L \in \mathcal{L}$;
3: **repeat**
4:    Find the root node $L_r$ and it's children $L_{child}$ be the largest set satisfing Proposition 2 and add the $L_r$ into $\mathbf{L}_r$;
5:    $\mathcal{L} = \mathcal{L} \setminus \{L_r \cup L_{child}\}$, $\mathcal{L}' = \{L_r \cup L_{child}\}$;
6:    **while** $\mathcal{L}' \neq \phi$ **do**
7:      Find the root node $L_r'$ from $\mathcal{L}'$ according to Proposition 2.
8:      $\mathcal{L}' = \mathcal{L}' \setminus \{L_r'\}$;
9:      Let $V_i, V_j$ be the children of $L_r'$;
10:       **for** each $L' \in \mathcal{L}'$ **do**
11:        $\mathcal{G} = \mathcal{G} \cup \{L_r' \to L'\}$;
12:        update $V_k$ (a child of $L'$) as $V_k = E_{(k,i|j)}$;
13:       **end for**
14:    **end while**
15: **until** $\mathcal{L} = \phi$
16: **if** $|\mathbf{L}_r| > 1$ **then**
17:    Construct an new latent variable $L$;
18:    $\mathcal{G} = \mathcal{G} \cup \{L \to L_r\}$ for all $L_r \in \mathbf{L}_r$;
19: **end if**
20: Remove the redundant edges of $\mathcal{G}$ using the method given in [Silva *et al.*, 2006]);
21: **Return:** $\mathcal{G}$

---

**Example 4.** *Continue to consider the example in Figure 1. Given the partial structure discovered in previous phase, i.e., $L_1 \to \{X_1, L_4\}$, $L_4 \to \{X_2, X_3\}$, $L_2 \to \{X_4, X_5\}$, $L_3 \to \{X_6, L_5\}$ and $L_5 \to \{X_7, X_8\}$, the algorithm proceeds is as follows. First, we find three latent variables $\{L_1, L_2, L_3\}$ in the partial graph $\mathcal{G}$ that cannot be further merged (Line 1). Second, we find that the latent variable $L_1$ is the root variable (Line 4). Third, we update data make use of $\{X_1, X_2\}$ (Line 12) and the results are given in Figure 2. Fourth, we find that $L_2$ a root latent variable of $L_3$ (Line 7), because $\{E_{(4,1|2)}, \{E_{(6,1|2)}\}$ and $X_5$ satisfies the Triad constraint, while $\{E_{(4,1|2)}, \{E_{(6,1|2)}\}$ and $X_7$ violates it. Finally, the whole structure is $L_1 \to \{L_4, L_2, L_3\}$, $L_2 \to L_3$, and $L_3 \to L_4$.*

## 5   Discussion of the Assumptions of Our Model

To understand the applicability of our model (Definition 1), we discuss the plausibility of the involved three assumptions and what may happen if they are violated.

If *Purity Assumption* is violated, i.e., there are directed links between observed variables, there may exist pure models equivalent to the underlying causal structure in terms of Triad constraints. For example, if we have enough data generated by the non-pure structure given in Figure 3, the estimated structure would be the one given in Figure 1. In the result, one essentially uses another latent variable (e.g., $L_4$) to replace the direct causal relation between the observed

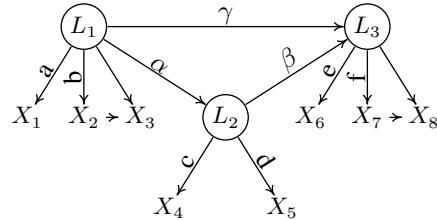

Figure 3: An non-pure latent causal structure, which can be transformed into the equivalent pure structure in Figure 1, by simply using a latent variable to represent the direct causal relation among the observed variables.

variables (e.g., $X_2$ and $X_3$). It is challenging but desirable to give a characterization of the result given by our procedure and its connection to the underlying causal structure in the general case.

For *Two-Pure Children Variable Assumption*, our assumption is much milder than that of Tetrad: we only need *two pure variables* for each latent variable, while Tetrad needs *three pure observed variables* for each latent variable. For *Non-Gaussianity Assumption*, we note that this assumption can be easily tested from the observed data. Furthermore, non-Gaussian distributions, unlike Gaussian ones, are expected to be ubiquitous, due to Cramér Decomposition Theorem [Cramér, 1962], as argued in Spirtes and Zhang [2016]. In fact, for our algorithm, this assumption can be relaxed to at most one noise term is Gaussian for observed variables, but not for latent confounders.

## 6  Simulation

For fair comparison, we simulate data following the linear latent structure model. There are four typical cases: Cases 1 and 2 have two latent variables $L_1$ and $L_2$, with $L_1 \rightarrow L_2$, and Cases 3 and 4 have three latent variables $L_1$, $L_2$, and $L_3$, with $L_2 \leftarrow L_1 \rightarrow L_3$, and $L_2 \rightarrow L_3$, respectively. Note that the simulated structure does not necessarily follow the pure assumption of our model (e.g. $X_2 \rightarrow X_5$ violates the purity assumption of our model), we simply recover the equivalent pure latent variable model for such structure as discussed in Section 5. In all four cases, the causal strength $b$ is sampled from a uniform distribution between $[-2, -0.5] \cup [0.5, 2]$, noise terms are generated as the fifth power of uniform(-1,1) variables, and the sample size is selected from $\{500, 1000, 2000\}$. The details of these networks are as follow.

- Case 1: $L_1$ and $L_2$ both have two pure measurement variables, i.e., $L_1 \rightarrow \{X_1, X_2\}$ and $L_2 \rightarrow \{X_3, X_4\}$.
- Case 2: adding impure variables to Case 1. We add $X_5$ and $X_6$ to $L_1$ and $L_2$ respectively, and add edges $\{X_2 \rightarrow X_5, X_4 \rightarrow X_6\}$.
- Case 3: each latent variable has two measurement variables, i.e., $L_1 \rightarrow \{X_1, X_2\}$, $L_2 \rightarrow \{X_3, X_4\}$, $L_3 \rightarrow \{X_5, X_6\}$.
- Case 4: adding impurities to Case 3. In detail, we add two measurement variables to each latent variable, i.e., add $X_7, X_8$ to $L_1$, $X_9, X_{10}$ to $L_2$, and $X_{11}, X_{12}$ to $L_3$. Further add edges $\{X_9 \rightarrow X_{10}, X_{11} \rightarrow X_{12}\}$.

Considering the data with non-Gaussian noise variables, we choose the Hilbert-Schmidt Independence Criterion (HSIC) test [Gretton *et al.*, 2008] as the independence test. We compared the proposed algorithm with the BPC [Silva *et al.*, 2006] and FOFC [Kummerfeld and Ramsey, 2016] algorithms[2]. The method by Shimizu *et al.* [2009] exploits BPC as its first step, so it is not used for comparison, given that BPC is included. All the following experimental results are based on 10 runs of the algorithms over randomly generated data.

In the experiment, the discovered measurement model and the reconstructed structure model are compared with ground truth to evaluate the performance of the algorithms. To evaluate the quality of the measurement model, we use *Latent omission*$=\frac{OL}{TL}$, *Latent commission*$=\frac{FL}{TL}$, and *Mismeasurement*$=\frac{MO}{TO}$ as the evaluation metrics, where $OL$ is the number of omission latent variables, $FL$ is the number of false latent variables, and $TL$ is the total number of latent variables in ground truth graph (See the details in [Silva *et al.*, 2006]) . To evaluate the quality of the reconstructed structure model, we further use the $F1 = \frac{2P \times R}{P+R}$ as our metric. Here $P$ and $R$ are the precision and recall, respectively.

As shown in Table 1, our algorithm, LSTC, achieves the best performance (the lowest errors) on all cases of the measurement model. Notably, when the sample size reaches 2000, the *latent omission, latent commission*, and *mismeasurements* of our method all reach 0. The BPC and FOFC algorithms (with the Delta test, a distribution-free test) do not perform well. These findings demonstrate that our algorithm requires only two pure variables in the measurement model, which is a clear advantage over the compared methods. Because of the clear performance gap, we only report the results of our methods on structure learning in Figure 4.

Table 1: Evaluation of output latent variables

| Algorithm | | Latent omission | | | Latent commission | | | Mismeasurements | | |
|---|---|---|---|---|---|---|---|---|---|---|
| | | LSTC | BPC | FOFC | LSTC | BPC | FOFC | LSTC | BPC | FOFC |
| *Case 1* | 500 | 0.00(0) | - | - | 0.00(0) | - | - | 0.00(0) | - | - |
| | 1000 | 0.00(0) | - | - | 0.00(0) | - | - | 0.00(0) | - | - |
| | 2000 | 0.00(0) | - | - | 0.00(0) | - | - | 0.00(0) | - | - |
| *Case 2* | 500 | 0.10(0) | 0.50(2) | 0.90(8) | 0.05(0) | 0.00(2) | 0.00(8) | 0.03(0) | 0.06(2) | 0.03(8) |
| | 1000 | 0.05(0) | 0.65(3) | - | 0.00(0) | 0.00(3) | - | 0.00(0) | 0.05(3) | - |
| | 2000 | 0.00(0) | - | - | 0.00(0) | - | - | 0.00(0) | - | - |
| *Case 3* | 500 | 0.20(0) | 0.86(6) | 0.96(9) | 0.03(0) | 0.00(6) | 0.00(9) | 0.17(0) | 0.71(6) | 0.93(9) |
| | 1000 | 0.13(0) | 0.93(8) | - | 0.00(0) | 0.00(8) | - | 0.00(0) | 0.85(8) | - |
| | 2000 | 0.00(0) | - | - | 0.00(0) | - | - | 0.00(0) | - | - |
| *Case 4* | 500 | 0.00(0) | 0.10(0) | 0.13(0) | 0.00(0) | 0.00(0) | 0.00(0) | 0.00(0) | 0.04(0) | 0.04(0) |
| | 1000 | 0.00(0) | 0.00(0) | 0.16(0) | 0.26(0) | 0.00(0) | 0.00(0) | 0.00(0) | 0.00(0) | 0.00(0) |
| | 2000 | 0.00(0) | 0.00(0) | 0.50(0) | 0.00(0) | 0.00(0) | 0.00(0) | 0.00(0) | 0.00(0) | 0.01(0) |

Note: The number in parentheses indicates the number of occurrences that the current algorithm *cannot* correctly solve the problem. If the result of a method is always wrong, we use the symbol '-' to indicate it.

As shown in Figure 4, the F1 score gradually increases to 1 as the sample size increases in all the four cases, which illustrates that our algorithm can recover the complete structure of the latent variables, including their causal directions.

# 7  Application to Stock Market Data

We now apply our algorithm to discover the causal network behind the Hong Kong stock market. The data set contains 1331 daily returns of 14 major stocks. Although some interesting results have been discovered on the data [Zhang and Chan, 2008], the latent variables behind the stocks are still unexplored.

The kernel width in the HSIC test [Gretton *et al.*, 2008] is set to 0.1. Note that the condition for finding clusters (Theorem 2) might be partially violated in the real world; we choose the candidate clusters with the highest number of satisfied Triad constraints in the algorithm, which proceeds as follows. First, $\{X_4, X_7, X_12\}$, $\{X_2, X_3, X_6\}$, $\{X_1, X_{10}, X_{11}\}$, $\{X_5, X_8, X_{13}\}$, and $\{X_9, X_{14}\}$ are identified as clusters by running the FindClusters algorithm. These five clusters are set to $L_2, L_3, L_4, L_5$

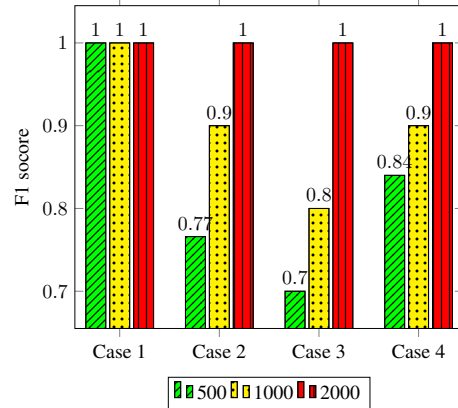

Figure 4: The F1 scores of LSTC algorithm.

and $L_6$, respectively. We then run algorithm 2 over the five clusters and obtain the final result, shown in Figure 5.

We have a number of observations from the discovered structure, which are consistent with our understanding of the stock market. 1) All stocks are affected by a major latent variable ($L_1$), which may be related to government policy, the total risk in the market, etc. 2) Companies in the same sub-index tend to gather under a common latent variable. For example, the cluster $\{X_5, X_8, X_{13}\}$ is in the Finance Sub-index; the cluster $\{X_2, X_3, X_6\}$ is in the Utilities Sub-index; the cluster $\{X_1, X_{10}, X_{11}\}$ is in the Properties Sub-index. 3) Ownership relations tend to have one common latent variable, i.e., $X_1$ holds about 50% of $X_{10}$, and they have one common cause $L_4$. Similarly, $X_5$ holds about 60% of $X_8$, and they have one common cause $L_5$.

# 8  Conclusion

In this paper, we proposed the so-called Triad constraints for estimating a particular type of linear non-Gaussian latent variable model. The constraints help locate latent variables and identify their causal structure. Then we apply these constraints to discover the whole structure of latent variables with a two-phase algorithm. Theoretical analysis showed asymptotic correctness of the proposed

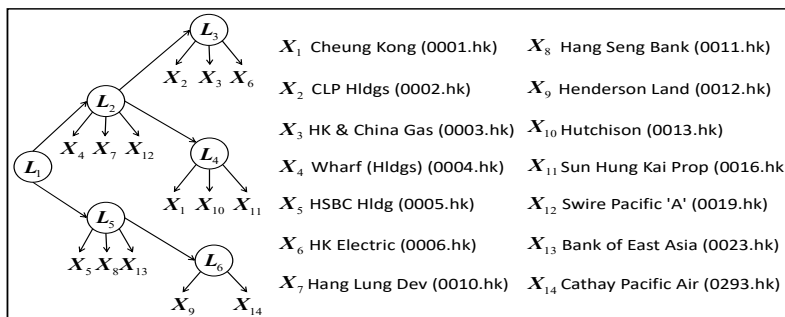

Figure 5: Causal diagram of the stocks

approach under our assumptions. Experimental results further verified the usefulness of our algorithm. Our future work is to 1) characterize properties of the results of our procedure for general causal structures with latent variables and 2) further relax our assumptions for better applicability of the method.

## Acknowledgments

This research was supported in part by NSFC-Guangdong Joint Found (U1501254), Natural Science Foundation of China (61876043), Natural Science Foundation of Guangdong (2014A030306004, 2014A030308008), Guangdong High-level Personnel of Special Support Program (2015TQ01X140), Science and Technology Planning Project of Guangzhou(201902010058) and Outstanding Young Scientific Research Talents International Cultivation Project Fund of Department of Education of Guangdong Province(40190001). KZ would like to acknowledge the support by NIH under Contract No. NIH-1R01EB022858-01, FAINR01EB022858, NIH-1R01LM012087, NIH-5U54HG008540-02, and FAINU54HG008540, by the United States Air Force under Contract No. FA8650-17-C-7715, and by NSF EAGER Grant No. IIS-1829681. The NIH, the U.S. Air Force, and the NSF are not responsible for the views reported here. KZ also benefited from funding from Living Analytics Research Center and Singapore Management University. Feng would like to thank Shohei Shimizu for his insightful discussions and suggestions on the original draft. We appreciate the comments from anonymous reviewers, which greatly helped to improve the paper.

## Footnotes

[2]We used these implementations in the TETRAD package, which can be downloaded at http://www.phil.cmu.edu/tetrad/.

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
