[Supplementary Material]

# Supplemental Material of "Triad Constraints for Learning Causal Structure of Latent Variables"

## A    Proof of the Theorems and Propositions

We begin with the Darmois-Skitovitch Theorem [Kagan *et al.*, 1973].
**Darmois-Skitovitch Theorem (D-S Theorem)**: Define two random variables $X_1$ and $X_2$ as linear combinations of independent random variables $n_i (i = 1, ..., q)$:

$$X_1 = \sum_{i=1}^{q} \alpha_i n_i, X_2 = \sum_{i=1}^{q} \beta_i n_i.$$

Then, if $X_1$ and $X_2$ are independent, all variables $n_j$ for which $\alpha_j \beta_j \neq 0$ are Gaussian. In other words, if there exists a non-Gaussian $n_j$ for which $\alpha_j \beta_j \neq 0$, $X_1$ and $X_2$ are dependent.

### A.1    Proof of the Theorem 1

**Theorem 1.** *Let $L_a$ and $L_b$ be two directed connected latent variables without confounders and let $\{X_i\}$ and $\{X_j, X_k\}$ be their children, respectively. Then if $\{X_i, X_j\}$ and $X_k$ violate the Triad constraint, $L_a \rightarrow L_b$ holds. In other words, if the Triad condition is violated and the latent variables have no confounders, then the latent variable of the reference variable is a child of the other latent variable.*

*Proof.* For $L_a$ and $L_b$, there are two possible causal relations, $L_a \rightarrow L_b$ and $L_b \rightarrow L_a$, corresponding to the causal structure (a) and (b) in Figure 1.

Figure 1: Identification of causal direction between two latent variables based on Triad constraints

In the non-trivial case, the causal strengths $\alpha, a, b$ and $c$ are not equal to 0.

As the variables strictly follow linear assumption, for the structure (a), we obtain

$$
\begin{aligned}
L_a &= \varepsilon_{L_a}, \\
L_b &= \alpha L_a + \varepsilon_{L_b} = \alpha \varepsilon_{L_a} + \varepsilon_{L_b}, \\
X_i &= a L_a + \varepsilon_{X_i} = a \varepsilon_{L_a} + \varepsilon_{X_i}, \\
X_j &= b L_b + \varepsilon_{X_j} = b \alpha \varepsilon_{L_a} + b \varepsilon_{L_b} + \varepsilon_{X_j}, \\
X_k &= c L_b + \varepsilon_{X_k} = c \alpha \varepsilon_{L_a} + c \varepsilon_{L_b} + \varepsilon_{X_k}.
\end{aligned}
\tag{1}
$$
$$
\tag{2}
$$

Then $E_{(i,j\,|\,k)}$ is as follows:

$$E_{(i,j\,|\,k)} = X_i - \frac{\text{Cov}(X_i, X_k)}{\text{Cov}(X_j, X_k)} \cdot X_j$$
$$= a\varepsilon_{L_a} + \varepsilon_{X_i} - t \cdot (b\alpha\varepsilon_{L_a} + b\varepsilon_{L_b} + \varepsilon_{X_j}),$$
$$= (a - \alpha tb)\varepsilon_{L_a} + \varepsilon_{X_i} - tb\varepsilon_{L_b} - t\varepsilon_{X_j}, \tag{3}$$

where $t = \frac{\alpha ac\text{Var}(\varepsilon_{L_a})}{(\alpha)^2 bc\text{Var}(\varepsilon_{L_a}) + bc\text{Var}(\varepsilon_{L_b})} \neq 0$.

For $\{X_i, X_j\}$ and $X_k$, based on Equation (3) and Equation (2), we can find that they have one common non-Gaussian noise $\varepsilon_{L_b}$ and $tb \neq 0, c \neq 0$. Hence, by D-S theorem, we have $E_{(i,j\,|\,k)} \not\perp\!\!\!\perp X_k$, i.e., $\{X_i, X_j\}$ and $\{X_k\}$ violate the Triad constraint.

Similarly for the structure (b), we have

$$L_b = \varepsilon_{L_b},$$
$$L_a = \alpha L_b + \varepsilon_{L_a} = \alpha\varepsilon_{L_b} + \varepsilon_{L_a}, \tag{4}$$
$$X_i = aL_a + \varepsilon_{X_i} = a\alpha\varepsilon_{L_b} + a\varepsilon_{L_a} + \varepsilon_{X_i},$$
$$X_j = bL_b + \varepsilon_{X_j} = b\varepsilon_{L_b} + \varepsilon_{X_j},$$
$$X_k = cL_b + \varepsilon_{X_k} = c\varepsilon_{L_b} + \varepsilon_{X_k}. \tag{5}$$

Then the pseudo residual $E_{(i,j\,|\,k)}$ is as below.

$$E_{(i,j\,|\,k)} = X_i - \frac{\text{Cov}(X_i, X_k)}{\text{Cov}(X_j, X_k)} \cdot X_j$$
$$= a\alpha\varepsilon_{L_b} + a\varepsilon_{L_a} + \varepsilon_{X_i} - \frac{a\alpha}{b} \cdot (b\varepsilon_{L_b} + \varepsilon_{X_j})$$
$$= a\varepsilon_{L_a} + \varepsilon_{X_i} - \frac{a\alpha}{b}\varepsilon_{X_j}. \tag{6}$$

For $\{X_i, X_j\}$ and $X_k$, based on Equation (6) and Equation (5), we find that there is no common non-Gaussian, independent component shared by $E_{(i,j\,|\,k)}$ and $X_k$. According to D-S Theorem, we reach the result that $E_{(i,j\,|\,k)} \perp\!\!\!\perp X_k$, i.e., $\{X_i, X_j\}$ and $\{X_k\}$ satisfy the Triad constraint. This finishes the proof. $\qquad\square$

## A.2 Proof of the Theorem 2

**Theorem 2.** *Let $S$ be a set of correlated variables. If $\forall X_i, X_j \in S$ and $\forall X_k \in \mathbf{X} \setminus S$, $\{X_i, X_j\}$ and $X_k$ satisfy the Triad constraints, then $S$ is a cluster.*

*Proof.* The proof is done by contradiction. Assume $S$ is not a cluster, the elements in $S$ must have at least two different parental latent variables. Without loss of generality, let $L_a$ and $L_b$ be the two latent variables, and let their children be $\{X_i, X_j\}$ and $\{X_k, X_l\}$, respectively. There are two cases to consider.

Case 1). When there is a causal relationship between $L_a$ and $L_b$, e.g., $L_a$ is the ancestor of $L_b$. We know that $X_k$ contains the noise $\varepsilon_{L_b}$ while $X_i$ and $X_j$ do not. Since $S$ is a correlated variable set, then $X_i$, $X_j$, and $X_k$ are related, i.e., $\text{Cov}(X_i, X_j) \neq 0$ and $\text{Cov}(X_k, X_j) \neq 0$. By $E_{(i,k\,|\,j)} = X_i - \frac{\text{Cov}(X_i, X_j)}{\text{Cov}(X_k, X_j)} \cdot X_k$, we obtain that $E_{(i,k\,|\,j)}$ must contain $\varepsilon_{L_b}$. According to D-S Theorem, $E_{(i,k\,|\,j)} \not\perp\!\!\!\perp X_j$, i.e., $\{X_i, X_k\}$ and $X_j$ violate Triad constraint, which contradicts the original assumption.

Case 2). When there is no causal relationship between $L_a$ and $L_b$. Since $S$ is a correlated variable set, we know $L_a \not\perp\!\!\!\perp L_b$. Therefore, $L_a$ and $L_b$ have at least one common ancestor. This structure shows that $\{X_i, X_j\}$ and $\{X_k, X_l\}$ contain respective latent variable noises, i.e., $\varepsilon_{L_a}$ and $\varepsilon_{L_b}$. Therefore, $E_{(i,k\,|\,j)}$ and $E_{(i,k\,|\,l)}$ both contain $\varepsilon_{L_a}$ and $\varepsilon_{L_b}$. Based on the D-S Theorem , $E_{(i,k\,|\,j)} \not\perp\!\!\!\perp X_j$ and $E_{(i,k\,|\,l)} \not\perp\!\!\!\perp X_l$, i.e., $\{X_i, X_k\}$ and $X_j$ violate Triad constraint, and so do $\{X_i, X_k\}$ and $X_l$. This contradicts the original assumption.

Based on the above analysis, Theorem 2 holds. $\qquad\square$

### A.3 Proof of the Proposition 1

**Proposition 1.** *Let $C_1$ and $C_2$ be two clusters. If $C_1$ and $C_2$ are overlapping, $C_1$ and $C_2$ share a same latent parent.*

*Proof.* Since $C_1$ and $C_2$ are two clusters, then the elements in $C_1$ have only one common latent variable. Without loss of generality, we let $L_1$ denote the parental latent variable of $C_1$. Similarly, $L_2$ denotes the parental latent variable of $C_2$. Since $C_1$ and $C_2$ are overlapping, then they have at least one shared element. Let $X_i$ denote the shared element of $C_1$. then $X_i$ has two latent parents $L_1$ and $L_2$, which contradicts with Theorem 1. This finishes the proof. $\square$

### A.4 Proof of the Proposition 2

**Proposition 2.** *Given a latent variable $L_r$ and its two children$\{V_i, V_j\}$, $L_r$ is a root latent variable if and only if $E_{(i,j|k)} \perp\!\!\!\perp V_k$ holds for all the $V_k$, where $V_k$ is a child of any other latent variables.*

*Proof.* (i) "$\Rightarrow$" Since $L_r$ is a root latent variable, there is no confounder between $L_r$ and another latent variables. Based on Theorem 1, we reach the conclusion that $E_{(i,j|k)} \perp\!\!\!\perp V_k$ holds for all $V_k$.

(ii) "$\Leftarrow$" This part of proof is proved by contradiction. Assume $L_r$ is not a root variable, $L_r$ has at least one parent. Considering the following two cases. Case 1: $L_r$ has only one parent. Let $L_s$ denote the parent of $L_r$ and $\{V_k\}$ denote a child of $L_s$. Based on Theorem 1, $E_{(i,k|j)}$ is not independent of $V_j$. Case 2: $L_r$ has more than one parent, e.g., $L_s$ and $L_t$, where their children variables are $\{V_k\}$ and $\{V_l\}$ respectively. If $L_s \perp\!\!\!\perp L_t$, based on the Theorem 1, one can find that $E_{(i,k|j)}$ is not independent of $V_j$ and that $E_{(i,l|j)}$ is not independent of $V_j$. If $L_s \not\perp\!\!\!\perp L_t$ and $L_s \rightarrow L_t$, due to linear assumption, $E_{(i,p|j)}$ always contains noise $\varepsilon_{L_t}$ and $V_p$ also contains this noise variable. According to D-S theorem, $E_{(i,p|j)}$ is not independent of $V_p$; Similarly, when $L_s \not\perp\!\!\!\perp L_t$ and $L_s \leftarrow L_t$, $E_{(i,k|j)}$ is not independent of $V_k$. The results of Case 1 and Case 2 show that the assumption does not hold, and $L_r$ is a root variable. This finishes the proof. $\square$

## B The correctness of Phase 2

We illustrate the correctness of Phase 2, especially the procedure of learning causal order of latent variables recursively, with the example in our paper.

Figure 2: The considerd structure in our paper, where (a) is the ground truth graph, (b) is the equivalent graph of (b)

In phase 1, we have get three clusters $\{\{X_1, L_4\}, \{X_4, X_5\}, \{X_6, L_5\}\}$ and their latent variables are $L_1$, $L_2$ and $L_3$, respectively. Next, we will show the process of Phase 2 step by step.

Note that, although we can not get the observed values of latent variables $L_4$ and $L_5$, we can use the values of their pure child as surrogates (This is because linear causal models are transitive). Here, we use $X_2$ and $X_7$ to replace $L_4$ and $L_5$, respectively, where $\varepsilon'_{X_2} = b\lambda\varepsilon_{L_4} + \varepsilon_{X_2}$ and $\varepsilon'_{X_7} = f\varepsilon_{L_5} + \varepsilon_{X_7}$ (See Figure 3).

We obtain,

$$X_1 = aL_1 + \varepsilon_{X_1},$$
$$X_2 = b\lambda L_1 + \varepsilon'_{X_2},$$
$$X_4 = cL_2 + \varepsilon_{X_4} = c(\alpha L_1 + \varepsilon_{L_2}) + \varepsilon_{X_4} = cL'_2 + c\alpha L_1 + \varepsilon_{X_4},$$
$$X_5 = dL_2 + \varepsilon_{X_5} = d(\alpha L_1 + \varepsilon_{L_2}) + \varepsilon_{X_5} = dL'_2 + d\alpha L_1 + \varepsilon_{X_5}, \qquad (7)$$
$$X_6 = eL_3 + \varepsilon_{X_6} = e((\gamma + \alpha\beta)L_1 + \beta\varepsilon_{L_2} + \varepsilon_{L_3}) + \varepsilon_{X_6}$$
$$= eL'_3 + e(\gamma + \alpha\beta)L_1 + \varepsilon_{X_6},$$
$$X_7 = f\theta L_3 + \varepsilon'_{X_7} = f\theta((\gamma + \alpha\beta)L_1 + \beta\varepsilon_{L_2} + \varepsilon_{L_3}) + \varepsilon'_{X_7}$$
$$= f\theta L'_3 + f\theta(\gamma + \alpha\beta)L_1 + \varepsilon'_{X_7}, \qquad (8)$$

We can then learn the causal structure of latent variable step by step in the following way.

- First, according to the Proposition 2, we know that $\{X_1, X_2\}$ correspond to the root latent cause.
- Next, we are ready to find the causal direction between the latent variables $L_2$ and $L_3$. Let us consider the pseudo residual variables $E_{(i,k|l)}$ instead of $X_i$, where $i \in \{4, 5, 6, 7\}$, $k \in \{1, 2\}$, $l \in \{1, 2\}$ and $k \neq l$. For convenience, we let $L'_2 := \varepsilon_{L_2}$ and $L'_3 := \beta\varepsilon_{L_2} + \varepsilon_{L_3}$. Then, we update the rest of variables by $\{X_1, X_2\}$,

$$E_{(4,1|2)} = X_4 - \frac{\mathrm{Cov}(X_4, X_2)}{\mathrm{Cov}(X_1, X_2)} \cdot X_1 = cL'_2 + \varepsilon_{X_4} - \frac{c\alpha}{a} \cdot \varepsilon_{X_1} \qquad (9)$$

$$E_{(6,1|2)} = e((\gamma + \alpha\beta)L_1 + \beta\varepsilon_{L_2} + \varepsilon_{L_3}) + \varepsilon_{X_6} - \frac{e((\gamma + \alpha\beta)}{a}(aL_1 + \varepsilon_{X_1}) \qquad (10)$$

$$= eL'_3 + \varepsilon_{X_6} - \frac{e((\gamma + \alpha\beta)}{a}\varepsilon_{X_1}.$$

- Finally, combining equations (7-10) and Theorem 1, one can see that $\{E_{(4,1|2)}, E_{(6,1|2)}\}$ **and $X_5$ satisfy the Triad constraint and that** $\{E_{(4,1|2)}, E_{(6,1|2)}\}$ **and $X_7$ violate it.** Figure 4 gives the graphical representations of the relationships among those variables, from which the above conclusion can be immediately seen. Thanks to this asymmetry, we know that the latent variable $L_2$, which generated $\{X_4, X_5\}$, is a cause of $L_3$, which generated $\{X_6, X_7\}$.

Figure 3: Using the Triad constraint to determine the direction between $L_2$ and $L_3$. (a) $\{E_{(4,1|2)}, E_{(6,1|2)}\}$ and $X_5$ satisfy the Triad constraint. (b) $\{E_{(4,1|2)}, E_{(6,1|2)}\}$ and $X_7$ violate the Triad constraint. The influences of noise terms are shown by dashed lines; note that the noise terms are mutually independent in each case.

# References

Abram M Kagan, Calyampudi Radhakrishna Rao, and Yurij Vladimirovich Linnik. Characterization problems in mathematical statistics. 1973.