[Reviews · NeurIPS 2019]

Reviewer 1



The authors focus on the challenge of causal discovery in the presence of unmeasured confounders. This is an important topic within the causal inference literature (and in fact many causal discovery algorithms often assume *no* unobserved confounders, which may often be unrealistic). Whilst some methods have been proposed, they often rely on assumptions such as pure 1-factor models with at least children per latent variable. The authors propose a two-stage method where they first use theorem 2 to find clusters based on whether triplets of variables satisfy a triad constraint. One important computational/algorithmic benefit of this first stage is that only independence testing (as opposed to conditional independence) testing is required. Given the clusters, the authors then focus on recovering the causal ordering over latent variables. The experiments are well executed. My only concern is the absence of traditional methods such as LiNGAM (even though it is misspecified). I would also have liked to see the performance of a very naive method which replaced the cluster finding using the triad constraints with simple clustering methods (eg k means clustering). This would help highlight which stage of the proposed method was really doing the heavy lifting (I would guess it is the first stage). Overall the paper is original and clearly written. There are some minor concerns regarding the experiments (some basic/misspecified baselines as discussed above would have been helpful). # minor comments/typos: - just before section 3: "equation 1)"

Reviewer 2



originality: the idea is very interesting, even though with heavy assumptions. Authors did explain the impact of each assumption, but it is still a very limited setting. quality: 1. the technical results are sound, but authors should state full assumptions for each theoretical results (such as Proposition 1). 2. One can view the work is closely related to some hierarchical tree/latent tree learning algorithm. It seems that the major different the latent variables can have arbitrary relationships. Author should explain in more details that how does the proposed algorithm compare with many latent tree algorithms? In Experiments, authors should also compare with these algorithms. 3. the consistency result of the algorithm is missing: is it sound or complete? 4. Does the method find an equivalent class of the graphs or the true graph? 5. what is a reason to choose noise term so small, with fifth power? It seems the algorithm could suffer from high noise? Clarity: the paper is well written, although one would wished that the authors should rely less on the supplementary materials and provide more intuition/explanation on proofs of the theorems. The examples are good. Significance: the idea is worth to pursue further and will have potential big impact. ===== I have read the authors' response. It would be interesting to see how the latent tree methods perform on the real dataset, since Figure 5 is basically a latent tree.

Reviewer 3



Update after author response: - Based on author response and some reflection on the problem itself (see below), I have increased my score to a 7. - Latent variable modeling is a challenging problem and any insights into additional constraints beyond standard conditional independence are valuable. The triad constraints mentioned here, while limited in scope, due to the parametric and structural assumptions posed in this paper, may be an interesting gateway to more generalized constraints in much the same way that tetrad constraints motivated this paper. - The notion of pseudo-residuals is also a concept worthy of further investigation, given its history in providing breakthroughs in other aspects of graphical modeling such as the Residual Iterative Conditional Fitting algorithm proposed in https://www.stat.washington.edu/~md5/Papers/2004uai.pdf. - If the paper gets accepted, I would ask that the authors change some of the language in the paper. Hyperbole such as "This goes far beyond the Tetrad constraints" can be off-putting and while Triad constraints are an improvement over Tetrad constraints, I am not sure they go "far beyond", or it should be left to the reader to decide if they do. ++++++++++++++++++++++++++++++++++++++++++++++ - The literature review in the introduction is very thorough! - "Overall, learning the structure of latent variables is still a challenging problem; for instance, none of the above methods is able to recover the causal structure as shown in Figure 1." Is the method proposed here, strictly more general than all of the other methods previously proposed. That is, are there a class of graphs that the present work would not be able to recover but previous methods would? My feeling is that it is strictly more general than those that use Tetrad constraints but it's unclear to me if it is more general than ICA-type methods like Hoyer et al (I don't think this is the case). - "It first finds pure clusters (clusters of variables having only one common latent variable and no observed parent) from observed data in phase I" -- this part of the methodology that requires a single latent common cause and no observed parents seems restrictive to me. - In definition 1 part 1), could you clarify what you mean by there is no direct causal relation between observed variables. Does this mean absence of a directed path meaning one cannot be an ancestor of the other or just absence of a directed edge meaning one cannot be a parent of the other. I interpret the current definition to mean the latter but if it is the former, this rules out important causal graphs such as the front-door graph A->M->Y, A<->Y (when viewing the latent projection) so being a little more explicit in the definition may be important. The former constraint however, is similar to the ancestrality property in ancestral graphs where the presence of both A <-> B and a directed path from A to B or vice versa is disallowed, and could be justified as such. - In definition 1 part 1), if my interpretation above for this part of the definition is correct i.e. there is no directed edge between the two observed variables, this is graphically equivalent to the absence of "bow-arcs" A->B, A<->B in the latent projection. This may affect the generalizability of the method beyond the current restrictions imposed on the latent structure (because latent projections define a class of infinite latent variable DAGs). The bow-free property combined with 2) and 3) are not sufficient for identifiability of all the parameters in linear SEMs with correlated errors. These graphs must lack C-trees or convergent arborescences in order to be everywhere identifiable. See [1]https://projecteuclid.org/download/pdfview_1/euclid.aos/1299680957. An example (I think) of a graph that fits Definition 1 but does not meet the criteria for identifiability is the following: A->B->C->D, A<->C, A<->D, B<->D. Since some of these parameters will correspond to coefficients or covariances involved in the computation of residuals, it seems like this would pose a challenge to generalizing this method further. - In definition 1 part 3) could it not be relaxed to at most one noise term is Gaussian? This would be similar to the assumption in other papers on causal discovery using additive noise models. Or do all noise terms have to be non-Gaussian for the DS-theorem? - In theorem 1: could you be precise, does "directed connected mean" existence of a directed path? Usually directed or bidirected connected implies the presence of a path but I think here it means L_a -> L_b or L_b -> L_a - The exposition in section 4.1 is nice and does a good job explaining the algorithm. - In section 4.1 -- regarding the replacement of the latent variable with an observed, does the correctness of this step rely on the linearity of the model i.e., collapsing directed paths is equivalent to multiplication/addition of coefficients as in circuit diagrams/path analysis (it kind of looks like that from some of the analysis in the supplement)? It might be useful for exposition to mention that if it is true. - In table 1, a comparison against an ICA-type method would be nice and would probably also the answer the question of how a triad based method compares with an ICA-type one. Minor comments: - Example 1 there is a typo -- vaiolated -> violated.

[Author Response · NeurIPS 2019]

We thanks the reviewers for the insightful comments and helpful suggestions. Please see below for our response.

**Reviewer 2**: Regarding the baseline methods in the experiments: Original LiNGAM assumes that there is no confounder. So the issue is that it is not clear how to compare its result with the groundtruth graph (with confounders). For LiNGAM with latent variables (LvLiGNAM) by Hoyer et al. (2008), the confounders are assumed to be independent, making it impossible to discovery their connectivity. Nevertheless, we include the results of LvLiNGAM for comparison (Table 1). For clustering methods, clearly, different assumptions for clustering will lead to different clustering results. K-means clusters are used to divide data points into groups, but in our case, we divide variables into groups.

**Reviewer 3**: 1. Regarding "the setting is limited": We totally agree, and at the same time would like to mention that linear latent variable models are common in the social sciences and ought to be more common in economics and elsewhere and that most, if not all, of the methods available for such problems have stronger (in one or another dimension) assumptions than the Triad method. 2. Regarding "compare with latent tree algorithm": Thanks for raising this issue. Generally speaking, they use the covariance information to recover the structure of the variables, ignoring non-Gaussinaity. Thus, our method can recover arbitrary DAG structure of latent variables, while the tree/latent tree learning algorithm can only solve the problems with the tree structure of latent variables the structures. We will include the results of three classic latent tree methods (neighbor-joining (NJ), by Saitou and Nei ( 1987), and recursive grouping (RG) and CLGrouping (CLNJ), by Choi et al. (2011)) in Table 1. 3. Regarding "the algorithm is sound or complete": The soundness and completeness of the algorithms were implied in the theoretical results, and will be made more explicit. In detail, Theorem 2 and Proposition 1 ensure the correctness of Phase 1 of our method (Algorithm 1), and Theorem 1 and Proposition 2 ensure the correctness of Phase 2 of our method (Algorithm 2). We will improve the presentation. 4. Regarding "find an equivalent class or the true graph": We are able to go beyond the equivalence class because of non-Gaussian of the data. we can uniquely recover the structure, including the structure over the latent variables, under our assumptions. 5. Regarding "noise with fifth power": Here, we set it to ensure the noise is clearly non-Gaussian. We also varied the powers and the results are included in the revised paper. Overall, we find that not surprisingly, the more non-Gaussian, the better the performance.

**Reviewer 4**: 1. Regarding "are there a class of graphs that the present work would not be able to recover but previous methods would?": There exist graphs following different assumptions from ours that can not recovered by our method. For instance, if the confounders are independent while the observe variables have directed edges in between, LvLiNGAM might be able to recover the graph, but our method cannot. However, under our model assumptions, there does not exist any graph that can be recovered by previous methods but not ours. 2. Regarding "unclear to me if it is more general than ICA-type methods": Yes, you are total right. It's not necessary more general than that–we allow direct causal relations between latent variables and rely on different assumptions. To the best of our knowledge, LvLiNGAM assumes that the latent variables are independent. 3. Regarding "require a single latent common cause and no observed parents": Here, our method can find the true graph under the purity assumption. As we discussed in our paper, if this assumption is violated, our method can still find an pure structure equivalent to the underlying causal structure. 4. Regarding "clarify there is no direct causal relation between observed variables": The latter is right, i.e., one cannot be a parent of the other. We will emphasize this point in the revision. 5. Regarding "generalizing this method further": Thanks for the interesting example. Definition 1 does not allow directed edges between two observed variables. We agree that it is nontrivial to generalize the method further, which we have been working on. 6. Regarding "could it not be relaxed to at most one noise term is Gaussian": Thanks for your insightful comments. Yes, this assumption can be relaxed to at most one noise term is Gaussian for observed variables, but not the latent variables. This can be seen from the proof, and will be discussed in the paper. 7. Regarding "directed connected mean existence of a directed path": Yes, it means directed path between $L_a$ and $L_b$ (there might be some intermediate variables in between). 8. Regarding the replacement of the latent variable with an observed: Yes, it relies on the linearity assumption. Following your suggestions, we will explain why in the paper. 9. Regarding "comparison against an ICA-type method": Thanks for the helpful suggestions. Please refer to the explanation in lines 3-7.

Table 1: Evaluation of output latent variables (due to space limitation, only results on case 1 and case 2 are reported )

| Algorithm | | Latent omission | | | | Latent commission | | | | Mismeasurements | | | |
|---|---|---|---|---|---|---|---|---|---|---|---|---|---|
| | | NJ | RG | CLNJ | LvLiNGAM | NJ | RG | CLNJ | LvLiNGAM | NJ | RG | CLNJ | LvLiNGAM |
| *Case 1* | 500 | 0.40(3) | 0.45(4) | 0.45(4) | - | 0.00(3) | 0.00(4) | 0.00(4) | - | 0.40(3) | 0.45(4) | 0.45(4) | - |
| | 1000 | 0.65(3) | 0.55(1) | 0.65(3) | - | 0.00(3) | 0.00(1) | 0.00(3) | - | 0.65(3) | 0.55(1) | 0.65(3) | - |
| | 2000 | 0.65(4) | 0.6(3) | 0.65(4) | - | 0.00(4) | 0.00(3) | 0.00(3) | - | 0.65(4) | 0.60(3) | 0.65(4) | - |
| *Case 2* | 500 | 0.35(2) | 0.50(4) | 0.40(3) | - | 0.00(2) | 0.05(4) | 0.10(3) | - | 0.46(2) | 0.58(4) | 0.53(3) | - |
| | 1000 | 0.55(3) | 0.65(3) | 0.60(3) | - | 0.00(3) | 0.00(3) | 0.00(3) | - | 0.70(3) | 0.77(3) | 0.73(3) | - |
| | 2000 | 0.75(7) | 0.80(7) | 0.75(7) | - | 0.00(7) | 0.00(7) | 0.00(7) | - | 0.83(7) | 0.9(7) | 0.83(7) | - |

[Meta-Review · NeurIPS 2019]

In this paper, the authors propose a novel kind of constraint they call the 'triad constraint' (presumably by analogy with tetrad constraints), which allow causal discovery to determine structure among latent variables, under certain (strong) parametric assumptions. The reviewers appreciated this paper as providing a novel contribution to the causal discovery literature.